

# From the index case to global spread: the global mobility based modelling of the COVID-19 pandemic implies higher infection rate and lower detection ratio than current estimates

Marian Siwiak, Pawel Szczesny and Marlena Siwiak

Data 3.0 Ltd., Sevenoaks, UK

Corresponding author
Marian Siwiak, siwiak@data30.co.uk

## ABSTRACT

**Background:** Since the outbreak of the COVID-19 pandemic, multiple efforts of modelling of the geo-temporal transmissibility of the virus have been undertaken, but none describes the pandemic spread at the global level. The aim of this research is to provide a high-resolution global model of the pandemic that overcomes the problem of biased country-level data on the number of infected cases. To achieve this we propose a novel SIR-type metapopulation transmission model and a set of analytically derived model parameters. We used them to perform a simulation of the disease spread with help of the Global Epidemic and Mobility (GLEAM) framework embedding actual population densities, commute patterns and long-range travel networks. The simulation starts on 17 November 2019 with the index case (presymptomatic, yet infectious) in Wuhan, China, and results in an accurate prediction of the number of diagnosed cases after 154 days in multiple countries across five continents. In addition, the model outcome shows high compliance with the results of a random screening test conducted on pregnant women in the New York area.

**Methods:** We have built a modified SIR metapopulation transmission model and parameterized it analytically either by setting the values of the parameters based on the literature, or by assuming their plausible values. We compared our results with the number of diagnosed cases in twenty selected countries, ones which provide reliable statistics but differ substantially in terms of strength and speed of undertaken Non-Drug Interventions. The obtained 95% confidence intervals for the predictions are in agreement with the empirical data.

**Results:** The parameters that successfully model the pandemic are: the basic reproduction number $R_0$, 4.4; a latent non-infectious period of 1.1. days followed by 4.6 days of the presymptomatic infectious period; the probability of developing severe symptoms, 0.01; the probability of being diagnosed when presenting severe symptoms of 0.6; the probability of diagnosis for cases with mild symptoms or asymptomatic, 0.001.

**Discussion:** Parameters that successfully reproduce the observed number of cases indicate that both $R_0$ and the prevalence of the virus might be underestimated. This is in concordance with the newest research on undocumented COVID-19 cases. Consequently, the actual mortality rate is putatively lower than estimated.

Confirmation of the pandemic characteristic by further refinement of the model and screening tests is crucial for developing an effective strategy for the global epidemiological crisis.

# INTRODUCTION

As of 23 April 2020, novel coronavirus SARS-CoV-2 has already spread into 185 countries and territories around the world (*Dong, Du & Gardner, 2020*). With over two and half a million confirmed infections and over 180 thousand deaths (*Dong, Du & Gardner, 2020*), it became a global challenge. COVID-19, the disease caused by this coronavirus, was characterized as a pandemic by WHO on 11 March 2020.

While several different measures to contain the virus have been implemented by countries all over the world, their effectiveness remains to be seen. Until an effective treatment is available, the accuracy of the pandemic models and the decisions made on their basis are the major factors in reducing the overall mortality in the COVID-19 pandemic.

The models used to inform decision-makers differ significantly in their basic assumptions because it is the first coronavirus of such an impact in terms of the number of fatal cases. Moreover, the existing modelling approaches often use biased data for tuning parameters or assessing models' quality. In particular, most models use country-level data which is biased by one or many of the following factors: (i) level of transparency in acquiring, analysing, interpreting and reporting of data, (ii) level of detection effort, (iii) efficiency of introduced Non-Drug Interventions, (iv) biased sampling of people to be tested (individuals showing severe and typical symptoms or suspected of having a contact with an infected person are more likely to be tested). The bias in data could be avoided by conducting well designed, random screening tests, but so far just a few such attempts have been made and they were limited to small and isolated communities in Italy (*Lavezzo et al., 2020*) and Germany (*Streeck et al., 2020*), or women admitted for delivery in New York (*Sutton et al., 2020*).

Also, as of the date of submitting this work, there were no peer-reviewed geo-temporal models of the pandemic. We argue that creating a global model by fitting curves to observed data is impossible, unless the data used in this exercise is a result of large scale screening tests. As mentioned above, country-level data is biased and the nature of this bias is different for each country, mainly reflecting state policy towards the disease. Obviously, fitting model curves on global data bears a significant error, as this data is a mixture of all country-level biases.

Even the proper characteristic of the virus is hampered by the above-mentioned biases. For instance, early estimates of the basic reproduction number of the virus, $R_0$,

were typically obtained using only Chinese data on the number of diagnosed cases (*Zhang et al., 2020*; *Wu, Leung & Leung, 2020*; *Liu et al., 2020*). These estimates proposed the value of $R_0$ within the range 1.5–6.47, and the earliest most likely served as the basis for the January's official WHO estimate, which stated $R_0$ = 1.4–2.5 (*World Health Organization (WHO), 2020a*). However, re-analysis of Chinese data provided an updated estimate of 5.7 (95% CI [3.8–8.9]) (*Sanche et al., 2020*).

Similar problems arise when estimating the actual prevalence of the virus. In this case, the estimates are not only biased by the state policy, but also by the fact that many infections are mild, asymptomatic or with atypical symptoms. In fact, many COVID-19 cases pass unnoticed, for instance in *Li et al. (2020a)* China study, over 50%.

The main objective of this research is to create a global model of the early stages of the pandemic that would overcome the problem of the biased data on the number of infected cases. This goal was achieved by creating the first global model of the COVID-19 pandemic that builds on top of the successful modelling framework Global Epidemic and Mobility (GLEAM) (*Van den Broeck et al., 2011*). In contrast to many existing models, our attempt did not use biased country-level data on the number of infected cases to fit the model curve. Instead, it used a set of predefined parameters to simulate the spread of the disease around the world starting from the index case placed in Wuhan, China on 17 November 2019. The exact date was suggested by unverified press reports and used widely as a date of a disease contraction for 'patient zero,' but evolutionary tracing also suggests a similar timepoint (*Li et al., 2020b*). The simulation took into account the current population densities all over the world, actual commute and flight networks, and travel ability of infected individuals. The simulation was run with 20 iterations for 154 days till 19 April 2020. The obtained results were used to create 95% confidence intervals for curves of cumulative number of diagnosed cases, separately for each country in the world.

The presented model enables better understanding of the virus, its infectivity and detectability. Also, it may serve as a solid foundation for further attempts of global and country-level modelling. In particular, more detailed models that include information on introduced precautions may be created by making the detectability parameter variable in time and geographics. This would enable an optimal pandemic strategy to be established for each country.

## MATERIALS AND METHODS

### Modelling software

The model is based on the GLEAM Model framework (*Balcan et al., 2010*), implemented in the GLEAMviz software (*Van den Broeck et al., 2011*). The GLEAM model integrates sociodemographic and population mobility data in a spatially structured stochastic disease approach to simulate the spread of epidemics at a worldwide scale. It was previously used for a real-time numerical forecast of the global spreading of Influenza A/H1N1 (*Tizzoni et al., 2012*), and the accuracy of that modelling was later confirmed (*Tizzoni et al., 2012*).

## Data sources

The reference data about the number of SARS-CoV-2 diagnosed patients in the period from 22 January 2020 to 19 April 2020, were downloaded from the Johns Hopkins University of Medicine Coronavirus Resource Center GitHub repository https://github.com/CSSEGISandData/COVID-19. The provided data have been grouped by country. These data sources were used to assess the quality of the model results. Empirical data for the time preceding 22 January 2020 is not available in the cited source.

Information about the severity of developed symptoms was derived from the worldometer.info website https://www.worldometers.info/coronavirus/.

Information on testing efforts in selected countries, comes from the https://ourworldindata.org/coronavirus-testing-source-data website.

Approximation of the number of mild and asymptomatic cases in the New York area was derived from the results of a random screening test performed on women admitted for delivery at the New York–Presbyterian Allen Hospital and Columbia University Irving Medical Center (*Sutton et al., 2020*).

Information on introduced Non-Drug Interventions comes from publicly available sources.

Other data sources, such as subpopulation selection, commuting patterns, or air travel flows, used during simulation, are embedded in the GLEAM software and well described by its developers (*Van den Broeck et al., 2011*).

## Model parametrization

Below and in Table 1, we present a set of parameters that was used in the model. Most parameters were derived from the literature. In the absence of a reliable reference, the parameters were assigned with the most plausible values by the authors based on the epidemiological knowledge on SARS-CoV-2 and other viruses. The parameters' derivation method is summarised in Table 1.

The average latency period (*lp*) of 5.6 days is a consensus of different estimations calculated previously (*Lauer et al., 2020*). Due to (1) long *lp*, effectively much longer than reported for other coronaviruses and (2) known cases of presymptomatic transmission (*Woelfel et al., 2020*; *Tong et al., 2020*), for the modelling purposes we decided to split the latency period into two parts: (1) average latent non-infectious period (*lnip*) of 1.1 days (based on the time of infectivity for other viruses (*Wallinga & Teunis, 2004*)) and (2) average presymptomatic infectious period (*pip*) of 4.5 days. This split produces two parameters used in the model:

(1) latency rate for the non-infectious period—non-infectious epsilon (*niε*):

$$ni\varepsilon \; = \; 1/lnip,$$

and

(2) latency rate for the infectious period—latency rate infectious epsilon (*iε*):

$$i\varepsilon \; = \; 1/(lp - lnip).$$
**Table 1  Analytically derived model parameters.**

| Parameter | Value | Description | Source/derivation |
|---|---|---|---|
| $R_0$ | 4.4 | Reproduction number for SARS-CoV-2 | Literature-based: assumed on the basis of infectivity rates of other coronaviruses. |
| $\mu$ | 7 days | Average recovery time since symptoms development | Literature-based |
| $\beta$ | 0.38261 | Transmission rate | $\mu \div R_0$ |
| $r\beta$ | 0.5 | Reduction in transmission rate resulting from the undiagnosed development of severe COVID-19 symptoms | Literature-based |
| $lp$ | 5.6 days | Average latency period | Literature-based |
| $lnip$ | 1.1 days | Average latent non-infectious period | Literature-based: assumed on the basis of non-infectious period of Influenza A/H1N1 |
| $pip$ | 4.5 days | Average presymptomatic infectious period | $lp - lnip$ |
| $ni\varepsilon$ | 0.9 (09) | Probability of transition from $lnip$ to $pip$ state | $1 \div lnip$ |
| $i\varepsilon$ | 0.2 (2) | Probability of transition from presymptomatic to symptomatic state | $1 \div (lp - lnip)$ |
| $pS$ | 0.01 | Probability of developing severe COVID-19 symptoms | Literature-data: The most often reported ratio of severe to mild symptoms |
| $pDS$ | 0.6 | Probability of being diagnosed when expressing severe COVID-19 symptoms | Assumed, taking into account that: 1. In elderly patients, COVID may be easily misdiagnosed, 2. Most of the countries in the world do not have sufficiently efficient healthcare systems |
| $tDR$ | 0.0061 | Rate of diagnosed SARS-CoV-2 infected individuals | Value representing the smallest possible detectability $> pS * pDS$ |
| $pDM$ | 0.00 (01) | Probability of being diagnosed when presenting mild or none COVID-19 symptoms | $(tDR - pS * pDS) \div (1 - pS)$ |

As the Republic of Korea provided high quality, reliable data and conducted a large number of tests during the pandemic, we decided to use Korean proportion of 'severe' to diagnosed cases as a base for the probability of developing the severe condition ($pS$), and we set it to 0.01. We assumed that patients with mild symptoms, in contrast to those in severe condition, are still capable of travelling. For model simplicity, we decided to merge into one compartment all mild and asymptomatic cases.

We decided to set the probability of detection of severe infection ($pDS$) to 0.6, in order to mimic two obstacles, typically preventing proper diagnosis. Firstly, the majority of patients with a severe course of the disease are either chronically ill or above 60 (*Zhou et al., 2020*)—their symptoms might be mistaken with those caused by their general health condition, and thus not reported on time. Secondly, the model is supposed to reflect the average illness detection around the globe, which includes many countries with low quality or underfinanced healthcare, where the number of SARS-CoV-2 tests is very limited.

Another parameter of the model, $pDM$ is the probability of being diagnosed with COVID-19 when having either mild symptoms or an asymptomatic illness course.

This parameter depends on previously defined $pS$ and $pDS$, as well as the ratio of total diagnosed to undiagnosed cases ($tDR$):

$$pDM \; = (tDR - pS * pDS) \; \div \; (1 - pS).$$

Knowing the limitations of previous modelling attempts (*Cowling et al., 2020*; *Ganyani et al., 2020*; *Zhang et al., 2020*; *Chen et al., 2020*; *Wu, Leung & Leung, 2020*; *Lin et al., 2020*; *Kucharski et al., 2020*), we decided to test a radically different COVID-19 epidemiologic paradigm, that is a significantly lower $tDR$. It means that in our model, we assume a higher proportion of undetected cases in comparison to other models proposed so far. Taking into account that none of them was capable of providing a plausible global simulation of the pandemic, plus the fact that the potentially low detectability has already been discussed in the literature (*Nishiura et al., 2020*; *Li et al., 2020a*; *Day, 2020a*, *2020b*; *Kimball et al., 2020*), we decided to test such a possibility in simulation by setting the lowest possible $tDR$. Its relation to $pDM$ sets its minimum to:

$$tDR \; > \; pS * pDS.$$

For previously set $pS$ and $pDS$ values, $tDR$ must be greater than 0.006, thus the value used in our simulation was set to 0.0061. This value means that for 10,000 of COVID-19 cases only 61 are positively diagnosed. The justification for such a strong assumption is based on the following: (i) $tDR$ reflects the average global detectability of the virus, including countries of low quality of public healthcare; (ii) $tDR$ reflects the average detectability of the virus during the entire examined period of 154 days that describes early stages of the pandemic; (iii) the percentage of asymptomatic or atypical cases is currently unknown, but small-scale screening tests conducted so far indicate that even 88% of examined diagnosed cases could be asymptomatic (*Sutton et al., 2020*); and (iv) some of the currently used tests might be faulty for example when viral load is small (*Pan et al., 2020*).

Other important and interconnected parameters required by the model are as follows: the effective contact rate, $\beta$; its reduction level for patients who developed severe symptoms of the disease but were not diagnosed, $r\beta$; and average recovery time since symptoms development, $\mu$. The parameter $\beta$ is derived from the time a host remains infectious, $d$, and the basic reproduction number of the virus, $R_0$:

$$\beta = R_0 \div d,$$

where:

$$d = \mu + pip.$$

The estimation of $R_0$ is a topic widely discussed in the literature, with values ranging from 1.4 to 6.49 (*World Health Organization (WHO), 2020b*; *Majumder & Mandl, 2020*; *Zhao et al., 2020*; *Imai et al., 2020*; *Read et al., 2020*; *Liu et al., 2020*). However, following the assumption of much higher than the currently suspected ratio of undiagnosed cases, we decided to use in our model a higher rate of transmissibility, yet well within the

range of 2–5, modelled for SARS (*Wallinga & Teunis, 2004*). The $R_0$ value assumed in our model is 4.4.

In our study μ is derived from a safe quarantine period for diagnosed cases (*Woelfel et al., 2020*). As the safe quarantine time is estimated to be 10 days (*Woelfel et al., 2020*), we assumed μ to last on average for 7 days from symptoms development to recovery. The sum of μ and previously estimated presymptomatic infectious period (*pip*) results in *d* equal to 11.5 days and β equal to 0.38261.

We decided to set *r*β to 0.5, following the assumption for this parameter used in GLEAM modelling of the 2009 influenza outbreak (*Balcan et al., 2010*). Patients who were diagnosed with COVID-19 are assumed isolated and as such, not spreading the disease any further.

## Model compartmentalization

To model the virus spread, we modified the compartmental SIR metapopulation transmission model to represent the nature of the COVID-19 epidemic.

In our model, we used seven different compartments (Fig. 1).

1. Susceptible population—equal to the general global population. We assume no existing immunity to infection.
2. Latent non-infectious—infected population in the first incubation stage, not yet infectious.
3. Presymptomatic infectious—infected population already infectious, but without developed symptoms.
4. Mild symptoms—joint populations of asymptomatic cases and those with inconspicuous symptoms.
5. Severe symptoms—population infected by SARS-CoV-2 with symptoms affecting their travel ability.
6. Diagnosed—population identified as infected with the SARS-CoV-2 virus. This is the reference line for the model accuracy.
7. Recovered—joint populations of recovered and fatal cases. We assumed recovered individuals cannot be reinfected throughout the simulation, although the only evidence so far is for *rhesus macaques* (*Ota, 2020*) and WHO is still investigating the issue. It shall be emphasized, that this assumption may be invalidated in the future, and any further utilisation of the presented model shall be accompanied by a thorough research into a current state of knowledge regarding the issue.

The last step was necessary to avoid the problem of unknown mortality ratio of the virus. It should be noted that the currently reported mortality ratio only applies to diagnosed cases (CFR, case fatality ratio), and its value still lacks consensus varying from 0.9% to 2.1% (*Wu et al., 2020*). The true mortality ratio (IFR, infected fatality ratio) that takes into account all undiagnosed cases is likely to be much smaller. For China for instance, it has been estimated at 0.66% (*Verity et al., 2020*). However, even if we assume that we currently detect all COVID-19 cases, it should not exceed 3.4% (WHO estimation) and this already is negligible from the perspective of our model (i.e. artificial increasing

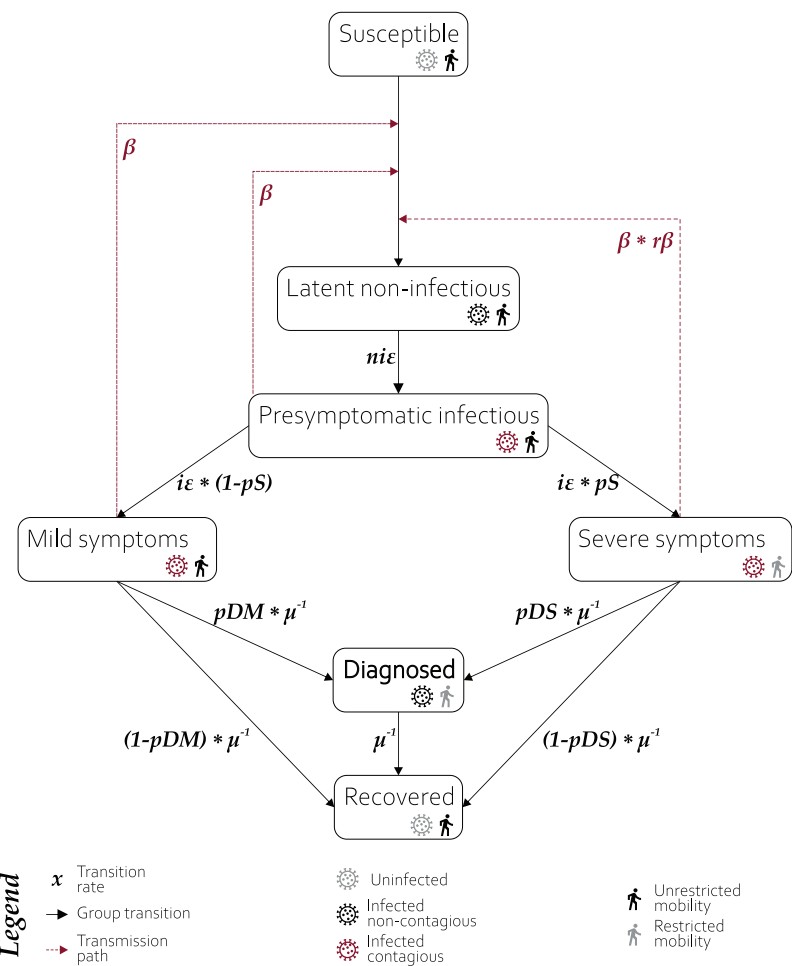

**Figure 1 Structure of compartments used in modelling.** A susceptible individual in contact with a person: (1) presymptomatic, (2) who developed mild symptoms, or (3) who developed severe symptoms, may contract the infection at rate $\beta$, $\beta$ or $r\beta*\beta$, respectively, and enter the latent non-infectious compartment where he is infected but not yet infectious. During the non-infectious period, each individual has a probability of $ni\varepsilon$ of becoming presymptomatic infectious. The presymptomatic cases have probability $i\varepsilon$ of developing severe or mild COVID-19 symptoms, with probabilities $pS$ and $1-pS$, respectively. The transition from both symptomatic groups occurs at $\mu^{-1}$ rate. Individuals who developed severe symptoms do not travel within and between modelled subpopulations and may be either diagnosed with probability $pDS$, or recover with probability of $1-pDS$. Individuals whose mild (or non-existent) symptoms are not stopping them from travelling may be diagnosed with probability $pDM$ or recover with probability $1-pDM$. The diagnosed individuals are considered isolated and effectively non-contagious and recover with rate $\mu^{-1}$. The recovery does not discriminate between true recovery and fatal cases.

of the number of recovered, and thus immune individuals by this amount should not affect the model's final outcome much).

## Model running

The prepared model served as an input for the GLEAM stochastic simulation. An initial simulation run with 10 iterations (the maximum allowed by publicly available free GLEAMviz Client application) was attempted prior to GLEAM development team

licencing a one-time temporary increase in the number of public version iterations to 20. The GLEAM framework uses high-resolution worldwide population data, allowing for the definition of subpopulations according to a Voronoi decomposition of the world surface centered on the locations of International Air Transport Association (IATA)-indexed airports (www.iata.org). Short-range commuting networks for the defined subpopulations are constructed on the basis of data on the commuting patterns of 29 countries in five continents, generalized to a general gravity law for commuting flows, reproducing commuting patterns worldwide.

The stochastic simulation of the pandemic was started on 17 November 2019. Although this date is stated only in non-academic sources, other reports also indicate mid-November as the time of the pandemic outbreak (*Li et al., 2020b*). The simulation began with a single presymptomatic individual located in Wuhan, China, and the development of the pandemic spread was modelled for 154 days.

### Model results processing

A single model run yields two sets of results. The first set is the median value and confidence intervals for the number of individuals per thousand which, at a given day, were moved to each of the compartments (presented in Fig. 1). The second set is the median value and confidence intervals for a cumulative number of individuals per thousand, entering each of the compartments, until a given day. Both sets of results can be extracted with different resolutions—globally, by hemisphere, continent, country, or the tessellated area surrounding IATA-registered airports.

For areas selected for detailed analyses of model results (i.e. twenty selected countries and the New York area), a cumulative number of transitions into a compartment of interest (i.e. 'Diagnosed' and 'Mild symptoms' respectively) was multiplied by the area population, and divided by thousand. The display of the model results was limited to the dates for which the experimentally derived data was available.

In order to compare model results with the random screening test from the New York area (*Sutton et al., 2020*), it was necessary to calculate the average number of undiagnosed mild and asymptomatic cases in this region ($c_s$) for the period covered by the experiment. The median and the lower and upper confidence limits of the number of individuals entering the 'Mild symptoms' compartment at any day ($n$) of the simulation is provided by the GLEAM framework ($c_{p+m}^n$, $c_{p+l}^n$ and $c_{p+u}^n$, respectively), while the number of individuals leaving the compartment ($c_{p-}^n$) was estimated similarly as in the framework:

$$c_{p-}^n = c_p^n \div \mu,$$

where $c_p^n$ stands for the average number of infected but not diagnosed asymptomatic or mild symptoms cases at day $n$. For each day ($n$) of the simulation, $c_p^n$ and its lower and upper confidence limits, $c_{pl}^n$ and $c_{pu}^n$, were calculated as:

$$c_p^n = c_p^{n-1} + c_{p+m}^n - c_{p-}^{n-1},$$
$$c_{pl}^n = c_p^{n-1} + c_{p+l}^n - c_{p-}^{n-1},$$
$$c_{pu}^n = c_p^{n-1} + c_{p+u}^n - c_{p-}^{n-1}.$$

The lower and upper confidence limits for $c_s$, respectively $c_{sl}$ and $c_{su}$, used for comparison with confidence interval derived from the screening test, were obtained as:

$c_{sl} = \left( \sum_{n=126}^{139} c_{pl}^n \right) / (139 - 126)$, and

$c_{su} = \left( \sum_{n=126}^{139} c_{pu}^n \right) / (139 - 126)$,

where $n = 126$ and $n = 139$ stand for simulation steps referring to 22 March 2020, and 2 April 2020, respectively.

The Excel workbook with performed calculations is provided as Supplemental Workbook 1.

## RESULTS

The simulation modelled the pandemic spread for 154 days. The results for all subpopulations around the globe are available in the shared model file (see the "Data Sharing" section below).

As overall data on the pandemic dynamics around the globe is likely to be biased by regions, often considerable in size and population, for which official statistics might be inaccurate, we decided not to compare overall model results with global data. Instead, we limited the analysis of results to 20 countries across five continents which are, in our belief: (a) divergent in the proportion of the tested population (as reported in https://ourworldindata.org/coronavirus-testing-source-data), quality of healthcare and strength of undertaken preventative measures; (b) likely to provide the public with real data. We also compared the model outcome for the New York area subpopulation with the results obtained in a random screening test on women admitted for delivery (*Sutton et al., 2020*).

The obtained 95% confidence intervals of predicted numbers of diagnosed patients were compared with empirical data from twenty countries. In Fig. 2, we present a percentage difference over time between the number of reported confirmed cases and confidence interval limits for modelled predictions. Positive values state that the model overestimates the number of diagnosed cases; negative values indicate the underestimations of the model; for the observed numbers of diagnosed cases that are within the model's CIs the percentage difference is equal to 0. For most of the selected countries, the model predictions fit well to the observed data, especially in the early stages of the pandemic.

Additionally, Fig. 3 confronts the number of actual confirmed COVID-19 cases with confidence intervals for the modelled number of diagnosed cases in twenty selected countries from five continents for all 154 days of the simulation. Some countries present epidemic dynamics different from the model. However, the direction of these deviations may be explained by the measures overtaken by their governments, their societal response, or the number of tests carried per million of citizens. To show the possible influence of Non-Drug Interventions, we summarised them and marked the dates of their introduction in the country charts. Even though it is difficult to assess the effectiveness of precautions without detailed reports from the country in question, in some cases they seem to explain the observed discrepancies well. For instance the number of detected cases in Japan, Australia and New Zealand is much smaller than predicted by the model, which

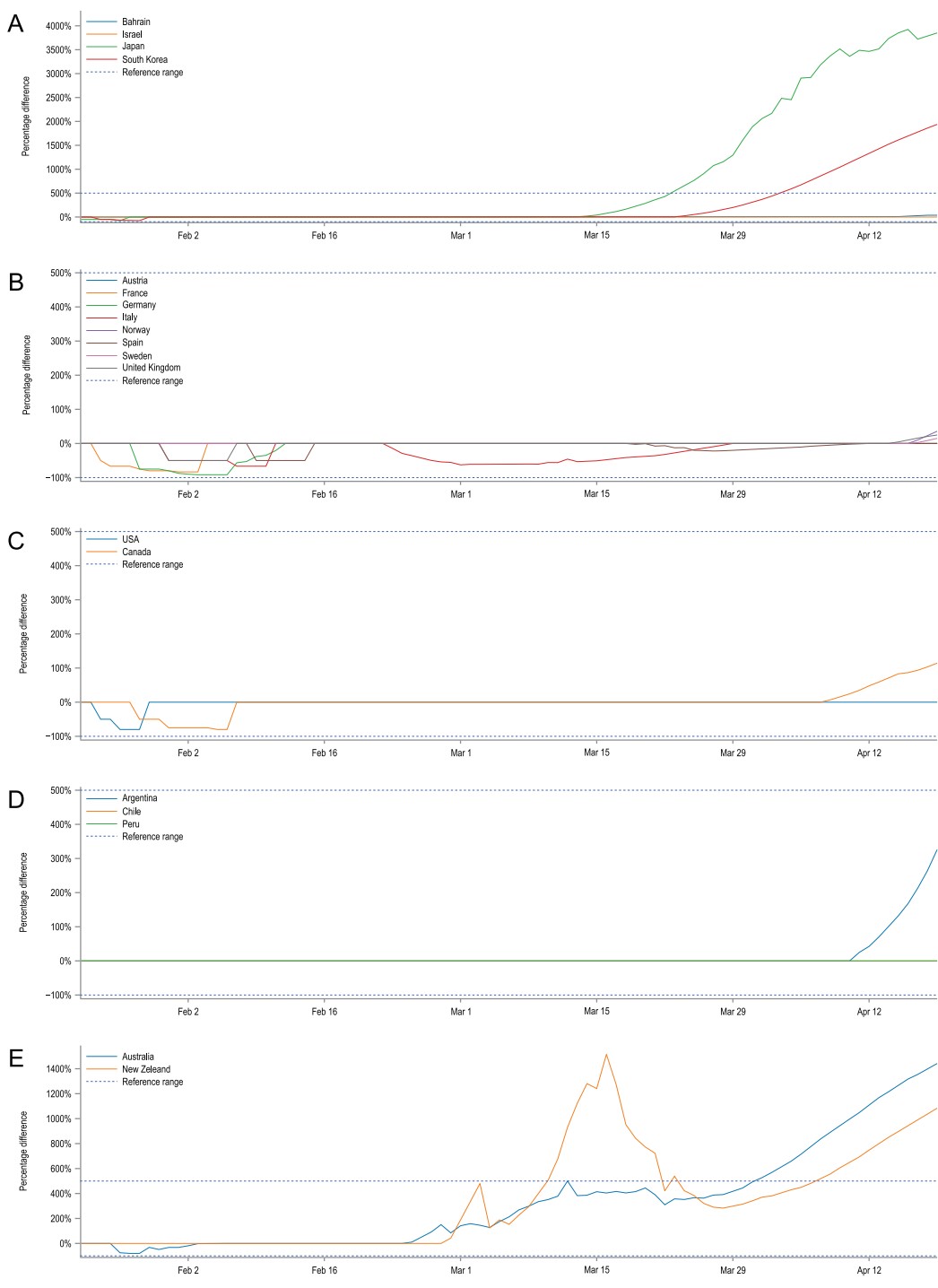

**Figure 2 Percentage difference over time between the number of reported cases and confidence intervals' limits for modelled predictions.** Positive values state that the model overestimates the number of diagnosed cases, negative values indicate the underestimations of the model. Observed numbers of cases that are within the model CIs are equal to 0. For clarity, country plots were grouped by continents and presented in five plots: (A) Asia, (B) Europe, (C) North America, (D) South America, (E) Australia and Oceania. The large discrepancies for Japan, Australia, New Zealand and the Republic of Korea are putatively caused by the fast and pronounced reaction of their governments and early introduced NDIs which are not reflected in our model.

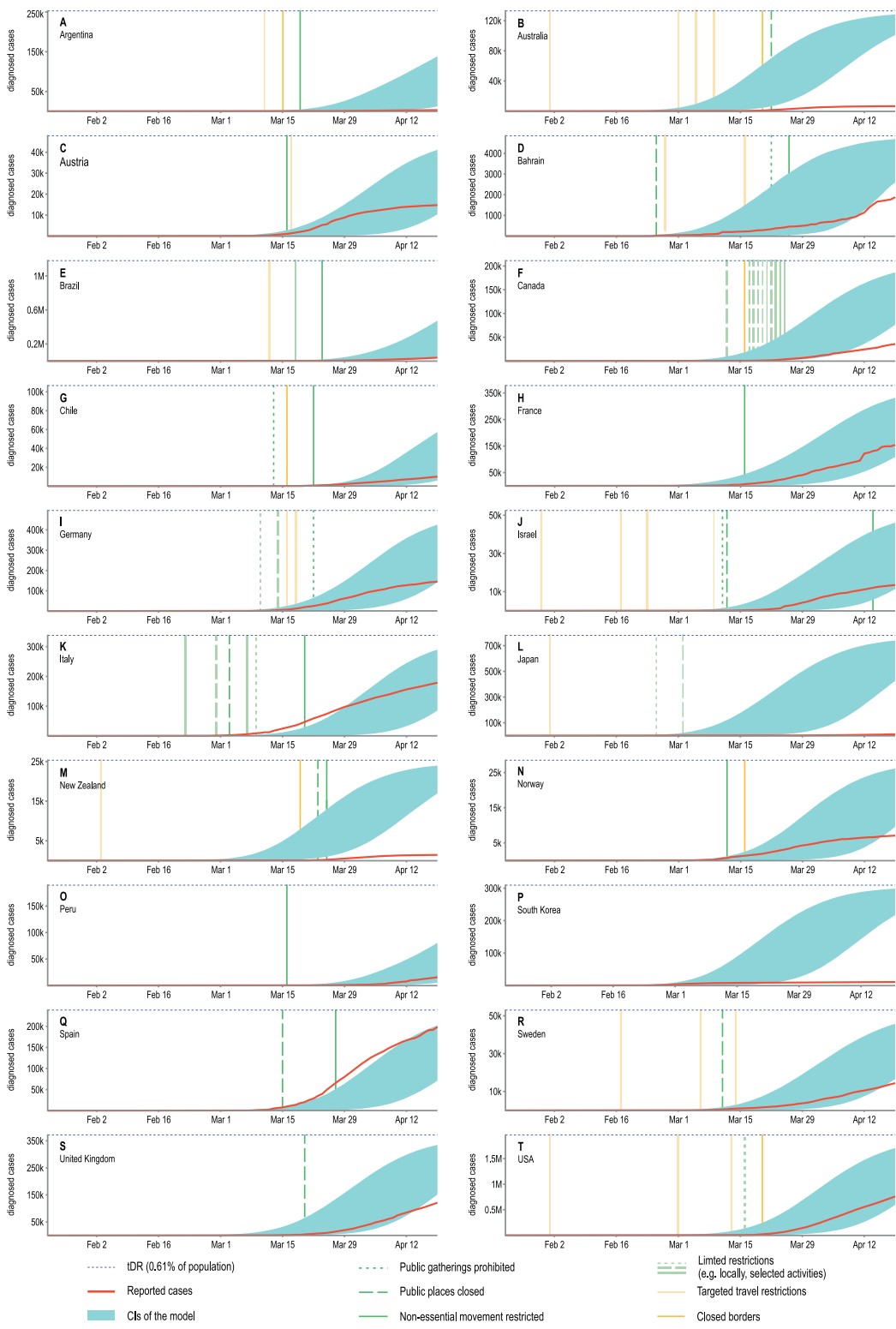

**Figure 3 An overlay of modelled confidence intervals for the predicted cumulative number of diagnosed cases and the actual reported values shown for twenty selected countries.** The *y*-axes show the absolute number of diagnosed cases and due to different country populations are not unified. To facilitate comparisons a blue, dotted line was added as a reference indicating the 0.61% of the total population of a country. This value is the same as the *tDR* parameter used in our model reflecting the

**Figure 3** (continued)
assumed ratio of detected to undetected cases. The confidence intervals obtained in our model will approach this value asymptotically. For most countries, observations agree well with model predictions. The observed discrepancies are most likely due to introduced NDIs which are not included in our model. The precautions were categorised and the dates of their introductions were marked on the plots with vertical lines, however, the assessment of their effectiveness is beyond the scope of this research. Analysed countries: (A) Argentina, (B) Australia, (C) Austria, (D) Bahrain, (E) Brazil, (F) Canada, (G) Chile, (H) France, (I) Germany, (J) Israel, (K) Italy, (L) Japan, (M) New Zealand, (N) Norway, (O) Peru, (P) South Korea, (Q) Spain, (R) Sweden, (S) United Kingdom, (T) USA.

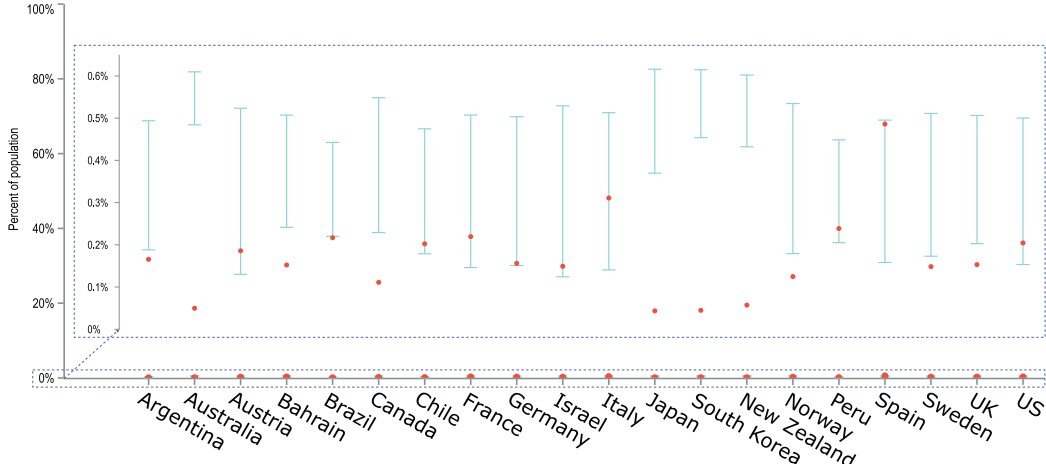

**Figure 4 An overlay of modelled confidence intervals with the empirical data on the number of diagnosed cases for the last day of the simulation, 19 April 2020.** The cumulative number of diagnosed cases is presented as a percentage of the country population, facilitating comparisons between countries. Model predictions are generally within the same order of magnitude as the observed data and obtained CIs are relatively narrow despite a limited number of iterations (20).

might be the result of the ban on flights from China introduced at the beginning of February combined with a geographical isolation of these countries. In contrast, most European countries started introducing preventive actions in March and the potential effects are only slightly visible in the last days of simulation. Similarly, the lack of concordance between model and empirical data in the case of the Republic of Korea may also be caused by the introduced precautions. However, in this case the preventive actions were of a different nature and are not shown in the country chart. Namely, the Republic of Korea introduced what was considered one of the largest and best-organised epidemic control programmes in the world, consisting of measures to screen and isolate any infected people, as well as track and quarantine those who contacted them (*South Korean Ministry of Health & Welfare, 2020*).

Furthermore, the empirical data from the last day of the simulation, 19 April 2020, was juxtaposed with the obtained confidence intervals in a single plot for all countries (Fig. 4). In this plot the number of detected cases is presented as a percentage of the country population, in order to better show the width of the obtained confidence intervals in

relation to the entire country population. From Fig. 4 it is visible that model predictions are generally within the same order of magnitude as the observed data. So are the differences between upper and lower confidence limits.

Finally, for the New York area and the time period between 22 March 2020 and 2 April 2020, we compared the model outcome with the results of a random screening test conducted on women admitted for delivery (Sutton et al., 2020). The cited experimental study revealed that out of 214 women tested, 33 were positively diagnosed with COVID-19, of which four had mild symptoms and 29 were asymptomatic. Assuming pregnant women are representative of the entire population and omitting severe COVID-19 cases from calculations (given patients in late pregnancy showing severe symptoms would have been admitted to the hospital earlier), the 95% CI for the true population proportion of mild or asymptomatic cases in the New York area, averaged over the examined period, is 0.11–0.21. According to our model, in the same area and averaged over the same period, the 95% CI for this proportion is 0.19–0.23. Although the predicted CI is not fully enclosed by the CI of the experimental result, their large overlap suggests high quality and accuracy of the model, started 139 days earlier from the index case on the other side of the globe.

Altogether, our 154-day long simulation of the pandemic seem to reflect the empirical data well. However, as is in the case of any model, this reflection is not perfect. The main reasons for the discrepancies between model predictions and the reported number of COVID-19 cases are the fast governmental responses and early introduced precautions, which significantly influence the pace of the disease spread. Such preventive measures, for instance local flight bans, are not included in our simulations. In fact, the model depicts only the 'natural' dynamics of the pandemic in the situation when governments do nothing to stop it. This means that in countries where overtaken actions were fast and effective, the model has a tendency to overestimate the number of detected cases.

The second potential reason for the observed discrepancies between model results and empirical data is the increase of the virus detectability in countries where the proportion of tested individuals is larger, leading to higher $tDR$ than the one assumed in our model. To check this hypothesis, we calculated 95% confidence intervals for the Spearman correlation coefficient between: (a) the cumulative number of conducted tests per capita in a country, and (b) the percentage difference between the cumulative number of detected cases and the lower or upper confidence limit of the CIs obtained in the model (i.e. if the model underestimates the number of detected cases, its upper confidence limit is used in calculations and vice versa; if the observed data is within the predictions of the model, the difference is zero). The correlations' CIs were calculated separately for each day of the simulation, if only sufficient data on countries' testing effort were available in the datasource. Missing data on the number of carried tests were interpolated if possible. As some countries started testing earlier than others, the number of datapoints for correlations varies from 4 to 18, depending on the date. The obtained CIs for correlations are plotted on a timeline in Fig. 5. It is visible that within the used data and with a limited sample size of 4–18 countries, it is not possible to state the direction of the correlation and decide if the tested hypothesis is true.

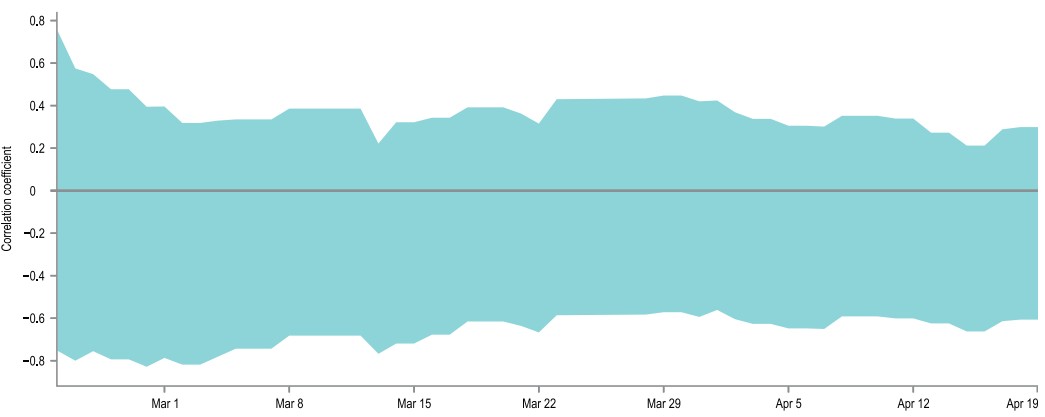

**Figure 5 Correlation between presented model accuracy and per-country testing effort.** An overlay of 95% confidence intervals for the Spearman correlation coefficients between the interpolated, cumulative number of conducted tests per capita in a country, and the percentage difference between the actual number of detected cases and the lower or upper confidence limit of the CIs obtained in the model. The correlations were calculated separately for each day of the simulation if only sufficient data on testing effort is available. As some countries started testing earlier than others, the number of data points for correlations varies from 4 to 18, depending on the date. The width of the obtained CIs and the values of their limits indicate that with the avaialble data it is not possible to state the direction of the analysed correlations.

We believe that further modelling efforts, that will include careful parameters' modifications over time in order to better reflect local responses to the pandemic, would greatly improve the accuracy of the simulations, but it is outside of the scope of this work.

## Data sharing

The model and the results of the simulation are freely available at https://github.com/freesci/covid19.

The data used to generate Fig. 5 is provided as Table S1.

The data and calculations used to obtain confidence intervals for the proportion of the mild and asymptomatic cases in the New York area is provided as Supplemental Workbook 1.

## DISCUSSION

To assess the quality of our model, its results should be compared to the observed data on the number of diagnosed cases, but this data suffers from four biases, listed previously. These biases cannot be totally eliminated, but careful selection of countries used in the analysis may reduce their impact significantly. In particular, we limited the analysis of model results to twenty countries which to our belief provide the most accurate and transparent reports on the number of infected cases (reducing bias nr 1). Selected countries are also divergent in terms of their detection efforts illustrated by the number of conducted tests per capita (reducing bias nr 2). Additionally, our model depicts the early stages of the pandemic when Non-Drug Interventions were not yet introduced on a large scale in the selected countries, and if they were, the exact time of intervention was added to the final diagrams to show its impact on the number of diagnosed cases (reducing bias nr 3). Furthermore, the proportion of symptomatic versus mild and

asymptomatic cases is built in the model, so is the fact that symptomatic individuals are more likely to be tested (reducing bias nr 4). Finally, the detectability of the disease is also built in the model. This means that the presented confidence intervals depict the plausible range of diagnosed cases assuming a given detectability, not the actual number of infected individuals in the country. However, the latter may easily be assessed knowing the assumed detectability, or derived from the model file provided with this article.

The presented model, due to its stochastic nature, avoids the problem of biased and inaccurate data and provides simulations of the pandemic spread for all the countries around the globe. It also has multiple implications concerning the major characteristics of the COVID-19 pandemic, such as the basic reproduction number of the virus $R_0$ (higher than previously assumed, yet not above the values estimated for other coronaviruses), and the average ratio of diagnosed cases $tDR$ (much lower than assumed so far, especially for cases expressing mild symptoms and asymptomatic). Such a low $tDR$ would indicate that the vast majority of the COVID-19 infections are so mild that they pass unnoticed. This is not implausible, considering the fact that there are 1.9 billion children aged below 15 years in the world (27% of the global population) and predominantly (ca. 90%) the course of their infections is mild or asymptomatic (*Dong et al., 2020*). Additionally, the $tDR$ used in our model indicates virus detectability averaged over the entire period of 154 days and over all countries in the world. Furthermore, some COVID-19 cases may show atypical symptoms (e.g. diarrhoea) (*Gao, Chen & Fang, 2020*) which hinder correct diagnosis. Taking all this into account, plus the results of our model and latest reports on the low detectability of the virus (*Nishiura et al., 2020*; *Li et al., 2020a*; *Day, 2020a*, *2020b*; *Kimball et al., 2020*), one may risk a hypothesis that the virus is already more prevalent in the global population than shown in official statistics at the moment, and consequently, its mortality ratio is much lower.

To verify this hypothesis, further actions are required. At first, the model should be refined by stochastic fitting of parameters to the observed data. Also the sensitivity analysis of the parameters should be performed. Such a refined model could be used for the analysis of the effectiveness of the applied Non-Drug Interventions and possibly, for the modelling of future NDIs strategies. Secondly, a simulation with the $tDR$ parameter increasing over time or diverging geographically might better reflect the actual virus detectability in the course of the pandemic. In this work, doubling the number of runs of simulation (from 10 to 20) did not influence the results in terms of their agreement with the observed number of cases. It allowed, however, an increase in the length of the simulation (from 135 to 154 days) while retaining the same order of magnitude of reported and predicted number of cases in each tested country. It means that in the case of extending the presented simulation further in time, it would be advised to introduce even a larger number of iterations to maintain the precision of predictions at the end of the simulation. Moreover, increasing the number of iterations would allow further narrowing of acquired CIs, increasing the model sensitivity to parameter values.

Finally, the real spread of the virus should be assessed empirically by conducting a sufficient number of tests on fully random samples as currently, most tests are limited to

individuals with strong and typical symptoms. Only after obtaining a solid measurement of the actual prevalence of the virus, can one draw conclusions about its true mortality rate.

We emphasise that our conclusions are a hypothesis based on a single computational model and without empirical verification, but they may serve as a platform for further research. At this stage, by no means should they be used as a reason for governmental decisions on lifting the precautions. Even if the true mortality of the virus is indeed lower than announced by the media, to our best knowledge many people remain in the high-risk group (e.g. elderly, chronically ill, etc. (*Baud et al., 2020*)). Lack of population resistance facilitates their contact with the virus and may lead to a rapid increase of severe cases in a short time, as seen for example in Italy (*Remuzzi & Remuzzi, 2020*), leading to the collapse of the healthcare system, which affects the entire society and results in many additional deaths not related to the virus itself. Careful use and tuning of Non-Drug Interventions, constant balancing of the disease spread and healthcare capacity, protecting the most vulnerable individuals, farsighted anticipation and agility in decision making may altogether be able to minimise the number of deaths without resulting in the global economic breakdown.

## CONCLUSIONS

Our model implies that the current consensus on the basic reproduction number of SARS-CoV-2 and its prevalence are misestimated. The overall global data on the pandemic dynamics seems strongly biased by large regions where official statistics may not reflect accurately the state of the epidemic, and by the fact that many COVID-19 cases may go unnoticed. The basic reproduction number of the virus should be confirmed based on reliable data, and its prevalence determined by conducting properly designed screening tests.

### Funding
The authors received no funding for this work.

### Competing Interests
Marlena Siwiak, Pawel Szczesny and Marian Siwiak are co-owners of Data 3.0 Ltd. Data 3.0 Ltd, however, does not operate in any capacity in the area of Life Sciences.

### Author Contributions
- Marian Siwiak conceived and designed the experiments, performed the experiments, analysed the data, prepared figures and/or tables, authored or reviewed drafts of the paper, and approved the final draft.
- Pawel Szczesny analysed the data, authored or reviewed drafts of the paper, provided biological context, and approved the final draft.
- Marlena Siwiak analysed the data, authored or reviewed drafts of the paper, provided statistical context, and approved the final draft.

## Data Availability

Files required to recreate the simulation, and full simulation results are available at GitHub: https://github.com/freesci/covid19.

## Supplemental Information

Supplemental information for this article can be found online at http://dx.doi.org/10.7717/peerj.9548#supplemental-information.

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
