# Peer review of "From the index case to global spread: the global mobility based modelling of the COVID-19 pandemic implies higher infection rate and lower detection ratio than current estimates"

_PeerJ, doi:10.7717/peerj.9548_

## Round 0.1 · original submission · Major Revisions

Dear Drs. Siwiak and colleagues:

Thanks for submitting your manuscript to PeerJ. I have now received two independent reviews of your work, and as you will see, one reviewer recommended rejection, while another suggested a major revision. I am affording you the option of revising your manuscript according to the two reviews but understand that your resubmission may be sent to at least one new reviewer for a fresh assessment (unless the reviewer recommending rejection is willing to re-review).

The reviewers raised many concerns about the manuscript. The manuscript lacks a clear delivery and needs to be restructured for clarity and effective delivery. The English needs substantial improvement. Many suggestions are provided by the reviewers, but you should consider an English expert to help with the revision.

There are several cases where claims are made without supporting references. Furthermore, your compartmental model equations need to be included, and a thorough explanation of how the GLEAM model weaves into the ord. diff. equations (or difference equations, please ensure that ODEs is correct here). It is also important to outline how your model differs from others (aside from being a posteriori model to fit the observed data).

Please consider contacting the developers of GLEAM and asking for use of more iterations. Both reviewers are struggling with the low number of iterations implemented and how removal of a single data point switches the direction of the association you have assumed. This is very troubling.

Please ensure that all of your methods provide the necessary information to be repeatable. Please ensure that all figures are clear and easy to read. There appear to be problems with the y-axis (different between countries) in certain figures.

Please avoid strong assumptions about infection frequencies and reporting by countries (e.g., Canada) in your revision.

Thus, I encourage you to revise your manuscript, accordingly, taking into account all of the concerns raised by both reviewers.

Good luck with your revision,

Best wishes,

-joe

Reviewer 1 ·

Basic reporting

The research utilized the Global Epidemic and Mobility Model (GLEaM) to in a posteriori approach to fit the model predictions to the observed data. Authors define the knowledge gap in COVID-19 existing models using biased data to fine tune model parameters, and I appreciate the authors stating that but it is not clear how their approach differs. This should have been made clearer to the reader, by specifically/stating why is their approach not biased compared to others who have fine-tuned their model parameters (as they did).
I also appreciate their attempt to describe findings however at times they are out of place, such as results presented in the intro. Also leaving out the objectives to be only implicitly presented rather than in a clear objectives statement, over use of the word “this” at start of many sentences which is confusing (which what?) should be reworded and addressed.

Experimental design

Lines 47-48 “Also, the existing modelling approaches often use biased data for tuning parameters or assessing models quality”
o Could you please add more explanation on how the existing models use biased data?
o Could you please explain how your model will avoid the problem of using biased data?
Lines 52-53” but none of the models succeeded to describe the pandemic at the global level”
o Could you please explain why?
o Please add examples and references.
o Could you please explain how your model will succeed in describing the global epidemic?
Line 60: Results presented in introduction please move to results section and continue your description from earlier introductory lines towards a clear knowledge gap that justifies your effort.
Lines 65-8: I appreciate the authors summarizing their report here but a statement of objectives of the modeling exercise is still needed
Materials and Methods:
What is the study population? There is no description of the study population? Is it only the 13 countries that you applied the model for?
Data sources:
Line 80 “The reference data about the number of SARS-CoV-2 diagnosed patients in the period from Jan 22, 2020, to Mar 26, 2020”
o Why you relied only on the data starting from Jan 22 to Mar 26, 2020 to build your model?
o What is your justification of using these data to simulate the global pandemic starting in November 12? As in lines 181-182.
Lines 87-89 Could you please briefly describe how the software adjusted for other data such as subpopulation selection, commuting patterns, or air travel flows?
Lines 91-92 “Below and in Table 1, we present two subsets of model parameters: 1) reliable and evidence-based derived from literature, and 2) knowledge and analysis-based estimations”
o In Table 1 there is only one set of parameterization could you please clarify the difference between both?
o Which one of the two sets provide the best outcome?
Line 124 “Knowing the limitations of previous modelling attempts” what are these limitations?
Line 121: You describe tDR as “rate of total diagnosed to undiagnosed cases”, this needs to be reworded as you state it’s a rate but define it as a ratio (use of “to”), and is it the rate of diagnosed only or both diagnosed and undiagnosed (omitting the word “to”)? Later in the discussion you describe it as rate of diagnosed cases (line 231), which is it?
Lines 134-5: The arithmetic dictates the inequality but why did you choose 0.0001 greater than 0.006, and shouldn’t this have been explored as a stochastic parameter?
Line 161: The model equations are missing, providing a third party input file is not sufficient. I am unable to reconcile your model specification. This results in several issues that I am unable to reconcile, such as, in your recovered (line 176) you state that it includes fatal cases. However, you don’t talk about the role of the recovered in reducing the number of susceptible in the population, if you are, then you also need to state whether you discounted the dead (fatal cases) since they wouldn’t impact herd immunity.
Line 177: This is not a sufficient number of iterations for a sensitivity analysis. I understand the limit by the software that would require a paid subscription, and I am not affiliated or have any conflict of interest here, but to base your entire paper on 10 iterations producing very little datapoints is not useful - case in point fig 2, there only 10 datapoints and removing the outlier on the far right may flip the association to positive instead of the authors interpretation of it being negative (Lines 211-2). Have you tried contacting the developers of GLEAM and perhaps they can allow them a limited period with more iterations.

Validity of the findings

Lines 187-188 ” we limited the analysis of results to thirteen countries across four continents”
o What is your justification? Could you please explain in more details with references?
Figure 2:
o Colors are confusing
o The confidence interval for the countries as in figures 4-16 has very wide range and by using this wide range the chances of your simulation to be in between is high. Could you please justify that?
Figure 3: why did you exclude Japan and Korea from the figure?
o what happens when you add both countries to the figure, does the correlation result differ. I would suggest reporting both values with and without Korea and Japan.
Line 236: The argument regarding the role of school students may not be valid given that most countries implemented that, or at least go back to the dates this was implemented and specify this in the model
Line 253-4: Explain what do you mean many people remain in the high risk? Provide your model data that supports this even if it is true.
General comment for the country figures need to be plotted in a single matrix (page size) as several plots in 1 figure and please unify the y-axis, as presented it projects a biased view of what is happening in different locations to the reader.

Lines 267-268 “Our model if confirmed, could be used as a tool for forecasting and optimizing non drug intervention and policy making” this conclusion is beyond the goals of the paper, please interpret your findings in terms of the issues with few iterations after clarifying your ordinary diff. equations, role of fatal cases being accounted when your model handles the susceptibles reduction over time (see earlier comment) and the posteriori approach of fitting model output to the observed data.

Minor comments:
Please add keywords for your manuscript
Line 23 “sixteen selected countries” while in line 187 “thirteen countries” which one is correct?
Line 71 “GLEaM” and line 72 “GLEAM” are they different terms?
Line 75 “global spreading of A/H1N1” do you mean Influenza A/H1N1.
Line 151=152: Move to results section
Line 192 “two countries which fulfil the above criteria” do you mean did not fulfil?
Line 254: Explain “as seen in Italy” replace with factual account, this may be known to most of us now but in 10 years of time new researchers will not be able to follow that without seeking other papers.

Reviewer 2 ·

Basic reporting

The language is OK, could definitely use some editing to improve the flow and clarity, references are not cited throughout the introduction section.

Experimental design

I struggled to locate a clear objective or research question, it appears that the authors are trying to tweak some of the model parameters and simulate the spread of pandemic so that they can mimic the case counts in some of the selected countries.

I have very serious concerns with some of the authors' approach:
1. It appears that the authors are suggesting that for every 610 diagnosed cases there are 100000 undiagnosed cases. That is a very strong assumption, there is definitely some underreporting of cases globally and a maximum of 1/3rd of cases could be asymptomatic but assuming that low diagnosed proportion without providing enough justification and supporting data may not be acceptable.
2. This is a stochastic model and the authors ran only 10 iterations of it to get the model outputs (Line 177). A stochastic model is not expected to give valid and meaningful insights with such a low number of model run. How did the authors conclude that 10 iteration was sufficent for model outputs to converge? If free version of the software was a limitation then that could not be acceptable for publishing a paper that may have significant impact on policy decisions during an ongoing pandemic.
3. On line 193 the authors state that Canada was excluded from the analysis because of its lack of coherency in reporting COVID-19 cases. I am not sure what the authors mean but this reviewer is not aware of any such incoherency in testing and reporting of COVID-19 in Canada.

Validity of the findings

It is hard to evaluate as the model have not been run for sufficient iterations and that is evident from very wide confidence intervals.

---

## Round 0.2 · Major Revisions

Dear Dr. Siwiak and colleagues:

Thanks for revising your manuscript based on the concerns raised by the reviewers. I have now received second reviews of your work from the original reviewers, and as you will see, both reviewers find the manuscript much improved. However, there are still concerns raised by both reviewers.

Please address these concerns ASAP. In particular, discuss why certain issues were not addressed in the prior round of revisions, as these are currently holding up acceptance of your work. PLEASE understand that a failure to address the original concerns of the reviewers this time will result in rejection of your work.

Good luck with your revision,

-joe

Reviewer 1 ·

Basic reporting

Thank you for addressing my comments in detail and for increasing the number of iterations, certainly better than the first simulation. The revised manuscript also brings more details justifying your assumptions and modeling approach. The figure edits are also very welcome.

Experimental design

The methods are much more clear now and the additional information certainly walked me through the authors approach in detail.

Validity of the findings

I do have a couple of minor suggestions:
1) Lines 130 to 163 in the tracked changes revised manuscript word document are a great discussion of your approach and limitations, hence this entire section should be weaved into the discussion. It really has no place in the introduction.
2) I am concerned about the assumption that recoverds do not contribute to infection any further, specially that some reports are coming out about limited duration of immunity with SARS and MERS lasted about 3 years. Its up to the authors but seems like out of caution they should choose to protect themselves by adding some verbiage in the discussion that clearly states that some aspects of the disease are still unknown.

Reviewer 2 ·

Basic reporting

The language of the manuscript has improved with the inclusion of citations, detailed description of sections, and presentation of a clear objective. However, this reviewer still finds the same issues with the manuscript and don't think that the authors had addressed those sufficiently.

1. The number of iterations: They increased the iteration from 10 to 20, both were arbitrary choices and not based upon any evidence that this number of iteration was sufficient for the model outputs to converge. They cite that the developers of the model felt that this number of iteration was sufficient; that is, unfortunately, no credible evidence. A stochastic model with so many sources of variation can not be expected to provide inferences with 20 iterations. The authors themselves acknowledge that "But we do not try to pretend our model is fully complete". It may have implications to publish a half baked science than the advantage it adds to scientific knowledge.

There is nothing novel in the manuscript that has mostly has not become common knowledge, except a set of parameters, that the authors are proposing, can mimic the number of cases in a few selected countries. To that end, the authors agree that any other potential combination of parameters may produce similar results suggesting such parameters may not reflect the reasonable estimates in the population.

Finally, the authors are using https://ourworldindata.org/ to obtain country-level data. The reviewer is not aware of the credibility of that source and finds it problematic that the authors without doing rigorous data retrieval efforts from individual countries appeared to slander some countries based on some global repository.

There was another minor issue that the reviewer noted, which is not using proper epidemiological terms: such as after pointing out by another reviewer the authors are still using the rate for mortality and detection which are not rates but a risk or ratio. The rate always has a time component. Similarly, the authors use "a single host" in their title, which also has an established term called "index case".

Experimental design

see above comments

Validity of the findings

see above comments

---

## Round 0.3 · Minor Revisions

Dear Dr. Siwiak and colleagues:

Thanks for revising your manuscript. The reviewer is very satisfied with your revision (as am I). Great! However, there are a few minor things to address. Please address these ASAP so we may move towards acceptance of your work.

Best,

-joe

Reviewer 1 ·

Basic reporting

The authors addressed my concern regarding discussion verbiage in the introduction.

Experimental design

The design is appropriate utilizing a modeling environment (Global Epidemic and Mobility (GLEAM) framework ).

Validity of the findings

Results are well described and authors did add the caveat statements regarding some of the model assumptions that future research may prove or disprove and that otherwise are currently acceptable.
I do understand the authors situation with the limited number of iterations, the costs required to run such proprietary modeling software etc. Also, I am glad the authors were able to go back to the software maker and gain more iterations but as I am sure the authors will agree, there is no doubt a larger number of iterations would have further increased the precision of their estimates by narrowing the 95% confidence intervals (CI) for each country's predictions. Perhaps a similar discussion point as the authors did for the permanent recovery assumption of their model, but for the number of iterations is due. I would even encourage the authors to state the reason, state:
In the methods: an initial simulation run with the trial version allowed X iterations was attempted prior to GLEAM software maker licensing a one time temporary increase in the trial version iterations to Y.
In the discussion: describe the changes or lack of in results due to increase in iterations from X to Y. Then discuss that it is expected that a larger number of iterations may have increased the precision of the country predictions, however, the 95% CI would have been expected to ever decrease in width given a larger number of iterations, indicating that the currently reported 95% CI may have coverage of the true predictions.

---

## Round 0.4 · accepted · Accept

Dear Dr. Siwiak and colleagues:

Thanks for revising your manuscript based on the concerns raised by the reviewers. I now believe that your manuscript is suitable for publication. Congratulations! I look forward to seeing this work in print, and I anticipate it being an important resource for groups studying the infection dynamics of COVID-19. Thanks again for choosing PeerJ to publish such important work.

Best,

-joe